# Chitosan-Based Antibacterial Films for Biomedical and Food Applications

**DOI:** 10.3390/ijms241310738

**Published:** 2023-06-27

**Authors:** Omar M. Khubiev, Anton R. Egorov, Anatoly A. Kirichuk, Victor N. Khrustalev, Alexander G. Tskhovrebov, Andreii S. Kritchenkov

**Affiliations:** 1Faculty of Science, Peoples’ Friendship University of Russia (RUDN University), Miklukho-Maklaya St. 6, 117198 Moscow, Russia; ihubievomar1@gmail.com (O.M.K.); sab.icex@mail.ru (A.R.E.); kirichuk-aa@rudn.ru (A.A.K.); alexander.tskhovrebov@gmail.com (A.G.T.); 2Zelinsky Institute of Organic Chemistry RAS, Leninsky Prosp. 47, 119991 Moscow, Russia; 3Institute of Technical Acoustics NAS of Belarus, Ludnikova Prosp. 13, 210009 Vitebsk, Belarus

**Keywords:** chitosan films, antibacterial properties, food industry applications, medical applications, nanocomposite materials

## Abstract

Antibacterial chitosan films, versatile and eco-friendly materials, have garnered significant attention in both the food industry and medicine due to their unique properties, including biodegradability, biocompatibility, and antimicrobial activity. This review delves into the various types of chitosan films and their distinct applications. The categories of films discussed span from pure chitosan films to those enhanced with additives such as metal nanoparticles, metal oxide nanoparticles, graphene, fullerene and its derivatives, and plant extracts. Each type of film is examined in terms of its synthesis methods and unique properties, establishing a clear understanding of its potential utility. In the food industry, these films have shown promise in extending shelf life and maintaining food quality. In the medical field, they have been utilized for wound dressings, drug delivery systems, and as antibacterial coatings for medical devices. The review further suggests that the incorporation of different additives can significantly enhance the antibacterial properties of chitosan films. While the potential of antibacterial chitosan films is vast, the review underscores the need for future research focused on optimizing synthesis methods, understanding structure-property relationships, and rigorous evaluation of safety, biocompatibility, and long-term stability in real-world applications.

## 1. Introduction

Chitosan (Figure 1) is a versatile and promising biopolymer that has garnered significant interest in various fields due to its unique characteristics and properties. Derived from chitin, which is predominantly found in the exoskeleton of crustaceans, insects, and the cell walls of fungi, chitosan is the second most abundant natural polysaccharide after cellulose [1,2,3]. Its biodegradability, biocompatibility, non-toxicity, antimicrobial properties, antitumor effect and ability to sensitize tumor therapy [4,5] make it an attractive material for numerous applications, particularly in the food industry and medicine [6,7,8,9].

One of the most noteworthy applications of chitosan is the development of antibacterial films. These films are utilized as food packaging materials, which can extend the shelf life of food products by inhibiting the growth of spoilage-causing microorganisms [10,11,12,13]. Additionally, chitosan-based films are employed in the medical field as wound dressings, where they can prevent bacterial infections and facilitate healing. Chitosan films also can be utilized as drug delivery systems, enabling the controlled release of antibiotics or other therapeutic agents [14,15,16,17].

In the current review, we will delve into the following aspects of chitosan-based antibacterial films:Structure and synthesis of chitosan: we will provide an overview of chitosan’s chemical structure, including its main functional groups and degree of polymerization. Additionally, we will discuss the various methods of chitosan production, such as the deacetylation of chitin and the use of specific enzymes or microorganisms;Antibacterial properties and mechanisms of antibacterial action of chitosan: we will explore the various factors that contribute to chitosan’s antimicrobial activity, such as its molecular weight, degree of deacetylation, and environmental conditions. Furthermore, we will discuss the proposed mechanisms of action through which chitosan exerts its antibacterial effects, including cell membrane disruption, chelation of essential nutrients, and interference with microbial gene expression;Types of antibacterial chitosan films: we will examine different types of chitosan films, such as those composed of pure chitosan, chitosan-metal nanoparticles, chitosan-metal oxide nanoparticles, chitosan-graphene, chitosan-fullerene or its derivatives, and chitosan-plant extracts. For each type of film, we will discuss their synthesis methods, unique properties, and potential applications in both the food industry and medicine;Applications in the food industry: we will explore how chitosan-based films can be used to improve food safety and quality by preventing microbial contamination, reducing oxidation, and enhancing mechanical and barrier properties;Applications in medicine: we will discuss the potential medical applications of chitosan films, including their use as wound dressings, drug delivery systems, and tissue engineering scaffolds.

The analysis of the current state in the field of chitosan-based antibacterial films and discuss potential future directions for further development of these materials are given in the sections that follow below.

## 2. Structure and Synthesis of Chitosan

Chitosan is a natural linear biopolymer derived from chitin, a major structural component found in the exoskeleton of crustaceans, insects, and the cell walls of fungi. Chemically, chitosan consists of randomly distributed β-(1→4)-linked N-acetyl-D-glucosamine (GlcNAc) and D-glucosamine (GlcN) units [18]. The proportion of GlcNAc and GlcN units defines the degree of deacetylation (DD), a critical parameter that influences the physical, chemical, and biological properties of chitosan, such as solubility, viscosity, and antimicrobial activity.

Chitosan’s molecular weight varies widely, depending on the source material and synthesis method. Higher molecular weight chitosan typically exhibits increased viscosity and mechanical strength, whereas lower molecular weight chitosan demonstrates enhanced solubility and bioactivity [19].

Producing chitosan from chitin involves various methods, primarily focusing on deacetylation processes:Alkaline deacetylation: this method is the most widely employed for chitosan synthesis. It involves treating chitin with concentrated alkali solutions, such as sodium hydroxide, at elevated temperatures. This process removes acetyl groups, converting chitin into chitosan. The reaction conditions, such as alkali concentration, temperature, and reaction time, influence the degree of deacetylation and the molecular weight of the resulting chitosan [20,21]. The undoubted advantage of alkaline deacetylation is its cheapness, while the main disadvantages of this method are associated with its environmental damage (the use of large amounts of aggressive reagents, i.e., alkalis);Enzymatic deacetylation: this method uses specific chitinase enzymes to selectively remove acetyl groups from chitin. The enzymatic process offers advantages such as specificity, milder reaction conditions, and reduced environmental impact compared to alkaline deacetylation. However, this method is less popular due to higher costs and lower yields. Ongoing research focuses on improving enzyme efficiency and lowering production costs [22];Microbial fermentation: this emerging method employs specific microorganisms, such as *Streptomyces* or *Bacillus* strains, to convert chitin into chitosan. The process holds the potential to be more sustainable and environmentally friendly compared to chemical methods. However, some challenges, including optimizing reaction conditions, scalability, and cost-effectiveness, remain to be addressed [23].

We want to point out that enzymatic deacetylation provides the highest degree of deacetylation (up to almost 100%) and preserves the chitosan backbone to the maximum. At the same time, for alkaline deacetylation, a degree of deacetylation of 10–20% is considered a great success. In addition, alkaline deacetylation leads to partial depolymerization of the polysaccharide [24]. Once chitosan is synthesized, it can be further processed to form films through various techniques (Figure 2):Solvent casting: the most common method for producing chitosan films involves dissolving chitosan in a suitable solvent (typically acetic acid), casting the solution onto a flat surface, and evaporating the solvent to form a solid film. This process results in films with good mechanical and barrier properties, making them suitable for food packaging and medical applications [25];Layer-by-layer assembly: this method involves the alternate deposition of oppositely charged polyelectrolytes (including chitosan) onto a substrate to produce multilayer films with controlled thickness and composition. The properties of these films, such as permeability, mechanical strength, and antimicrobial activity, can be tailored by adjusting the number of layers, the choice of polyelectrolytes, and the assembly conditions. Layer-by-layer assembled chitosan films have been used in applications such as food packaging with controlled release of antimicrobial agents and medical devices with tunable drug release profiles [26,27].

In the following sections, we will discuss the antibacterial properties of chitosan, the underlying mechanisms of its antibacterial action, and the various types of antibacterial chitosan films, along with their applications in the food industry and medicine.

## 3. Antibacterial Properties and Mechanism of Action of Chitosan

Chitosan, a natural linear biopolymer derived from chitin, has gained considerable attention due to its broad-spectrum antimicrobial activity against various microorganisms, including Gram-positive and Gram-negative bacteria, fungi, and yeasts [21]. This activity is influenced by several factors, such as molecular weight, degree of deacetylation (DD), pH, and the ionic strength of the environment.

The molecular weight of chitosan plays a critical role in determining its antimicrobial properties. Lower molecular weight chitosan tends to exhibit higher antimicrobial activity due to its increased solubility, which allows for better penetration of bacterial cell walls [19]. The DD of chitosan also significantly impacts its antimicrobial activity, with higher DD chitosan showing enhanced activity because of the increased density of protonated amino groups that interact with bacterial cell surfaces [28].

The antimicrobial action of chitosan involves several mechanisms, which can be summarized as follows (Figure 3):Cell membrane disruption: the cationic nature of chitosan, resulting from the protonation of its amino groups under acidic conditions, enables it to interact with the negatively charged bacterial cell surface. This interaction can cause changes in the cell membrane permeability, leading to the leakage of intracellular components, disruption of membrane integrity, and eventual cell death [29];Chelation of essential nutrients: chitosan possesses the ability to bind to essential metal ions such as calcium, magnesium, and iron, which are crucial for bacterial growth and metabolism. By chelating these essential nutrients, chitosan can inhibit bacterial growth and biofilm formation, impeding the ability of bacteria to colonize surfaces and cause infections [30];Interference with microbial gene expression: Chitosan can penetrate bacterial cells and interact with intracellular components such as DNA and RNA. This interaction can lead to the inhibition of bacterial gene expression and protein synthesis, ultimately resulting in bacterial growth inhibition and cell death [31];Reactive oxygen species (ROS) generation: Chitosan has been reported to induce the generation of ROS in bacterial cells, leading to oxidative stress, DNA damage, and cell death [32]. This mechanism contributes to the antimicrobial activity of chitosan and its derivatives.

In the upcoming sections, we will explore various types of antibacterial chitosan films, including pure chitosan films, chitosan films with metal nanoparticles, chitosan films with metal oxide nanoparticles, chitosan films with graphene, chitosan films with fullerene derivatives, and chitosan films with plant extracts. For each type of film, we will discuss their synthesis methods, unique properties, and potential applications in both the food industry and medicine.

## 4. Antibacterial Films from Pure Chitosan

Pure chitosan films have been extensively studied due to their inherent antimicrobial properties, biocompatibility, and biodegradability. These films can be used in various applications, such as food packaging, wound dressings, and medical device coatings. In this section, we will discuss the synthesis and properties of pure chitosan films, along with their applications in the food industry and medicine.

### 4.1. Use in the Food Industry

Chitosan films have been widely investigated for food packaging applications to prolong shelf life, maintain quality, and reduce spoilage caused by microbial contamination. These films can be synthesized using various methods, including solvent casting, electrospinning, and layer-by-layer assembly, as mentioned earlier. It should be noted that, due to the extreme rigidity of the chitosan macromolecule, the addition of a plasticizer is required to obtain a film with good mechanical properties. As a rule, glycerin is used as a plasticizer [33].

Chitosan films have been employed to preserve different types of food products, such as fruits, vegetables, meats, and fish. For example, chitosan films have been used to extend the shelf life of strawberries by reducing weight loss, maintaining firmness, and inhibiting fungal growth [34]. Similarly, chitosan-coated chicken breasts have shown reduced bacterial growth and lower lipid oxidation rates compared to uncoated samples [35]. It has also been found that chitosan films have similar oxygen permeability values to the commercially available food packaging ethylene-vinyl alcohol copolymer films or polyvinylidene chloride films [36].

### 4.2. Use in Medicine

In the medical field, pure chitosan films have been employed for wound dressings, drug delivery systems, and medical device coatings due to their antimicrobial, biocompatible, and biodegradable properties [24].

Chitosan films can promote wound healing by providing a moist environment, preventing bacterial infections, and stimulating tissue regeneration. These films have been shown to accelerate wound closure, reduce inflammation, and enhance re-epithelialization in animal models [37]. Chitosan films can be used for the improvement of the Schwann cell response in nerve regeneration [38], for the tissue engineering of bone, supporting the bone growing in the attaching parts as well as preserving the integrity of the structure during in vivo remodeling of tissue [39]. Chitosan-based dressings have been extensively applied for the preparation of wound healing materials due to their excellent properties, such as good gas permeation, high porosity, and high surface area. Exudate removal, moisture retention, skin regeneration, cell respiration, and hemostasis were all aided by using chitosan films [40]. Yang et al. developed a film-forming solution for the treatment of MRSA infections in wounds. The film-forming solution was more suitable for the treatment of MRSA than a wide range of previously described systems based on synthetic polymers [41].

Chitosan films can also potentially be used as drug delivery and sustained release systems. The release of a drug from chitosan films is followed by polymer degradation and a complex diffusion process. Drug release behavior can be influenced by a variety of factors, including drug state, surface functionalization, and polymer properties [42,43]. Chitosan-based films can be used for the transdermal delivery of proteins and peptides [44] and antibiotics [45].

The main disadvantages of films based on pure chitosan are associated with their meager mechanical properties and moderate antibacterial activity. The most important way to expand the mechanical and antibacterial properties of chitosan films is the introduction of various active additives into the polymer matrix of the film. In the following sections, we will discuss various types of antibacterial chitosan films, including chitosan films with metal nanoparticles, chitosan films with metal oxide nanoparticles, graphene, fullerene and its derivatives, and also chitosan-based films with plant extracts. For each type of film, we will discuss their preparation methods and potential applications in both the food industry and medicine.

## 5. Antibacterial Films from Chitosan with Metal Nanoparticles

The incorporation of metal nanoparticles into chitosan films can enhance their antimicrobial properties, making them more effective in various applications. In this section, we will discuss the synthesis and properties of chitosan films with metal nanoparticles, such as silver, gold, and copper, and their applications in the food industry and medicine.

### 5.1. Use in the Food Industry

Chitosan films containing metal nanoparticles can be synthesized using methods like in situ reduction, electrospinning, and solvent casting. Among the different metal nanoparticles, silver nanoparticles (AgNPs) have been widely studied due to their strong antimicrobial activity against a broad spectrum of microorganisms [46].

Chitosan films incorporating AgNPs have been used in food packaging to prolong shelf life and maintain quality. For example, chitosan films containing AgNPs have been shown to effectively inhibit the growth of spoilage bacteria on chicken breast fillets, extending their shelf life [47]. As compared to the chitosan film, various properties were enhanced in the chitosan/AgNPs blend films, including light barrier, opacity, elongation at break, as well as bioactivities, thus suggesting that films could be used as novel alternative food packaging application [48]. Chitosan films containing covalent organic frameworks (COFs) immobilized AgNPs showed that the tensile strength of the nanocomposite films enhanced dramatically with the increase of the COFs-AgNPs content, while the UV–visible light barrier property, water swelling and solubility properties, and water vapor permeability decreased significantly. Furthermore, the CS/COFs-AgNPs nanocomposite films also showed outstanding antibacterial activity and effectively prolonged the storage time of white crucian carp (*Carassius auratus*) [49]. Recently, Padil et al. demonstrated that the addition of AgNPs to the polymer blend matrix results in significant improvement of the mechanical characteristics of chitosan-based films. The cross-linking motion of the silver nanoparticles leads to a decrease in swelling degree, moisture retention capability, and water vapor permeability. The addition of 0.0075% of the nanoparticles dramatically increases the tensile strength of the films. The real-time application of the films was tested by evaluating the shelf-life existence of carrot pieces covered with the composite films. The composite film containing AgNPs becomes effective in lowering bacterial contamination when compared to plastic polyethylene films [50].

Similarly, chitosan films with gold nanoparticles (AuNPs) and copper nanoparticles (CuNPs) have also demonstrated enhanced antimicrobial activity and potential applications in food packaging [51] Chitosan-based films with copper nanoparticles are characterized by enhanced antimicrobial activity against *Salmonella* and *S. typhimurium* as well as improved mechanical behavior in comparison with the corresponding pure chitosan films [52]. Cardenas et al. obtained a film based on chitosan and colloidal copper nanoparticles by casting with microwave heating. The film-forming solution has good dispersion and film-forming properties due to the high density of amino and hydroxyl groups of chitosan, which avoids the aggregation of metal particles. The elaborate composite film has distinct advantages over chitosan film in terms of oxygen permeability, vapor permeability and light transmission. In addition, the resulting film is characterized by a high antimicrobial effect and can be used to protect food products and extend their shelf life [53].

### 5.2. Use in Medicine

Chitosan-based films reinforced by metal nanoparticles are increasingly being used in the biomedical field. This is mainly due to the antibacterial effect of chitosan films, which in many cases is significantly enhanced by the introduction of metal nanoparticles into the polymer matrix [54]. In addition, in some instances, metal nanoparticles and chitosan macromolecules have a symbiotic positive effect on the stimulation of tissue growth and regeneration, which is important for wound healing [55].

Research has demonstrated that chitosan films imbued with AgNPs foster the wound recovery process by creating a damp milieu, lessening inflammatory responses, and invigorating the renewal of impaired tissues [56]. These films also exhibit a protective quality against wound contaminations, effectively halting the proliferation of harmful microbes, including *Staphylococcus aureus*, *Pseudomonas aeruginosa* [56] and *E. Coli* in vitro and in vivo [57]. Recently elaborated chitosan films with silver nanoparticles demonstrated high antibacterial activity against *E. coli.* At the same time, the prepared multilayers showed mild activity against *S. aureus* predominantly due to the antiadhesive effect, and this material looks promising as a prospective material for covering the surface of medical implants in order to reduce their bacterial contamination [58]. Chitosan-based dressing hydrogel films with AgNPs, elaborated Karami et al. showed strong antibacterial activity against *Bacillus cereus* and *Staphylococcus aureus* and a positive effect on tissue regeneration [59].

In drug delivery systems, chitosan films containing metal nanoparticles can be used as carriers for various bioactive agents, including antibiotics, anti-inflammatory drugs, and growth factors. The release of these agents can be controlled by modifying the film’s composition, thickness, and degree of cross-linking [60].

Chitosan films with metal nanoparticles have also been used as coatings for medical devices, such as catheters, implants, and prosthetics, to prevent bacterial adhesion and biofilm formation. These coatings can minimize the risk of device-related infections, which are a significant concern in healthcare settings [61].

## 6. Antibacterial Films from Chitosan with Metal Oxide Nanoparticles

The incorporation of metal oxide nanoparticles into chitosan films can also improve their antimicrobial properties and broaden their applications. In this section, we will discuss the synthesis and properties of chitosan films with metal oxide nanoparticles, such as zinc oxide (ZnO) and titanium dioxide (TiO_2_) as the most used oxides in chitosan films, and their applications in the food industry and medicine.

### 6.1. Use in the Food Industry

Chitosan films containing metal oxide nanoparticles can be synthesized using methods like in situ oxide generation, direct introduction of metal oxide into a polymer matrix, electrospinning, and solvent casting. Zinc oxide nanoparticles (ZnONPs) have been widely investigated due to their antimicrobial properties against a broad range of microorganisms [62].

Chitosan films incorporating ZnONPs have been used in food packaging to extend shelf life and maintain quality. For example, chitosan films containing ZnONPs have been shown to inhibit the growth of spoilage bacteria on fresh-cut fruits, such as melons and apples [63]. Chitosan and zinc oxide nanoparticles loaded gallic-acid films showed good physical properties such as oxygen and water vapor permeability, swelling, UV-vis light transmittance and significant antibacterial potential [64]. Hamideskar et al. obtained a composite film based on chitosan as a polymer matrix and zinc oxide nanoparticles as a filler. The nanocomposite had a good porous structure with interconnection. Evaluation of the antibacterial activity of the produced films showed that the synergistic effect between chitosan and zinc oxide improves the antibacterial activity of the nanocomposite. The potential of the nanocomposite for chicken fillet and cheese packaging has been explored. Microbiological analysis of food samples during storage showed that the most effective inhibition of inoculating bacteria was obtained for chicken fillet and cheese samples wrapped with a nanocomposite film. The use of the film protected the physicochemical quality of the chicken fillet and cheese samples during storage. At the same time, the weight loss of chicken fillet and cheese samples wrapped with a nanocomposite was lower than that of samples wrapped with pure chitosan or even polyethylene film. Therefore, antibacterial nanocomposite film is suitable for poultry and cheese packaging, which holds great promise for improving the quality and shelf life of the mentioned food products [65]. Chitosan-ZnO coating prepared by Nabulsi et al. demonstrated a reduction in the initial count of *E. coli* during 28 days with a constant dispersal of nanoparticles within the polymer medium. The elaborated coatings showed excellent effectiveness for white-brined cheese shelf-life prolongation [66]. Similarly, chitosan films with titanium dioxide nanoparticles (TiO_2_NPs) have also demonstrated enhanced antimicrobial activity and potential applications in food packaging [67]. Chitosan-alginate nanocomposite film with nanoparticles of TiO_2_ showed enhanced tensile strength and elongation at break as well as antimicrobial activity against foodborne pathogens *E. coli*, *S. aureus*, *S. typhi*, and *L. monocytogene*. The film with 0.1% of TiO_2_ demonstrated the complete killing of gram-positive bacteria. Moreover, the film was completely biodegraded during the 3 months [68]. The composite membrane of chitosan and TiO_2_NPs enhanced the firmness of the mango. The peroxidase and polyphenol oxidase activity of the composite coated fruits, as well as the total phenol and flavonoid content was also higher than in control group, that indicated that the chitosan/nano-TiO_2_ composite coating could maintain the nutrient composition of mangoes [69]. Mannaa et al. stated that TiO_2_ incorporated in chitosan film enhanced it mechanical properties and antimicrobial activity compared to blank film, that suggest they could be a viable alternative to non-biodegradable food-packaging material [70].

### 6.2. Use in Medicine

In medical applications, chitosan films with metal oxide nanoparticles have been used mainly for wound dressings. Chitosan films with ZnONPs have been shown to promote wound healing by providing a moist environment, reducing inflammation, and stimulating the regeneration of damaged tissue [62]. In addition, zinc oxide has a pronounced antibacterial effect, including against many strains of microorganisms that are resistant to conventional antibiotics [71]. Recently, Teplea et al. prepared novel chitosan-based films with ZnO nanoparticles for wound healing. The film has significant activity against *S. aureus* and *C. albicans* being promising materials for antimicrobial wound dressing, especially to protect against infections. The authors suppose that symbatic action of chitosan and ZnO nanoparticles is beneficial in healing and tissue regeneration [72]. A similar film with zinc oxide nanoparticles was elaborated by Sodagar, who noted the excellent antibacterial effect of the film on *Streptococcus mutans*, *Streptococcus sanguis*, and *Lactobacillus acidophilus* and demonstrated its potential for the treatment of diseases of the oral cavity (dentistry) [73].

## 7. Antibacterial Films from Chitosan with Graphene

Graphene is an allotrope of carbon consisting of a single layer of atoms arranged in a hexagonal lattice nanostructure [74]. Incorporating graphene into chitosan films can enhance their antibacterial properties and expand their applications. First, graphene is an extremely strong material, and it greatly enhances the strength of chitosan films [75]. It is also worth noting that graphene has the biocompatibility with mammalian cells required for use as a scaffold structure for tissue engineering [76,77].

### 7.1. Use in the Food Industry

The introduction of certain amounts of graphene into the polymer matrix of chitosan films can significantly improve their physicochemical and biological properties, thereby expanding the scope of these films [78]. Chitosan films incorporating graphene have been used in food packaging to extend shelf life and maintain quality. Recently, high-performance graphene-containing films based on chitosan have been developed for potential applications such as food packaging. The introduction of small amounts of graphene oxide significantly increases the Young’s modulus, as well as the elasticity of the obtained films; in addition, the introduction of the mentioned carbon nanostructures significantly improves the barrier properties of the material. It should also be noted the high antibacterial activity of chitosan-graphene films in comparison with blank chitosan films [79]. These studies are in agreement with previous data on the antibacterial activity of graphene [80].

### 7.2. Use in Medicine

And yet, most of the examples described in the literature of the use of chitosan films with graphene developed for the biomedical field. In particular, graphene-containing systems are proposed as film-like scaffolds and coatings for tissue engineering [81,82,83], systems with enhanced loading capacity and releasing profile for drug delivery [84,85], antibacterial films and hydrogel films for wound healing [86,87].

The most widespread application of graphene-containing films based on chitosan concerns drug delivery. In these cases, graphene structures often act as the core, which is the main force in absorbing the pharmacological substance or genetic material via π–π stacking interactions, hydrophobic interactions, van der Waals forces, and in some instances, hydrogen bonds [88]. Chitosan plays the role of a polymer matrix, which is characterized by biocompatibility, biodegradability and non-toxicity. In addition, the characteristic features of the polymer matrix also largely determine the kinetics of drug release [89].

## 8. Antibacterial Films from Chitosan with Fullerene and Its Derivatives

Fullerene, a type of carbon substance characterized by its substantial electron density and extensive volumetric surface area, shows immense potential. Its bonding energy is somewhat low, generally resulting in physisorption. This structure offers a platform for diverse modifications, potential electrical uses, transporting interstitial atoms, as well as functioning as a substance for adsorption and storage. There are two known forms of adsorption in fullerene: one weaker form on its upper surface and another stronger one in the gaps between its spherical molecules. Fullerene’s uses are numerous, including serving as an antiviral, an antioxidant and neuroprotectant, an X-ray contrast agent, an antimicrobial, a tool in genetic manipulation, an implant prosthesis, and a diagnostic and treatment agent for oral cancer [90,91]. Incorporating fullerene and its derivatives into chitosan films is often able to give the material new properties. In this section, we will discuss the synthesis and properties of chitosan films with fullerene and its derivatives and their applications in the food industry and medicine.

Fullerene’s unique properties also include antimicrobial activity against a variety of microorganisms [92]. In addition, fullerene reinforcement of chitosan films can be used to improve their strength, elasticity, and barrier characteristics [93]. All these parameters of food films and coatings are of paramount importance. In particular, the introduction of C_60_ fullerene in the chitosan matrix enhances the thermal, viscoelastic, and optical properties of the biodegradable chitosan-based composite film [94]. The authors of this paper recommend this elaborated film for use in food biotechnology. Similarly, there are a few examples of the introduction of fullerenol into the polymer matrix of chitosan films. Fullerenol-60 is a water-soluble hydroxylic derivative of C_60_ fullerene, i.e., C_60_(OH)_24_ [95]. Fullerenol in chitosan-based films potentiates the antibacterial effect of chitosan and promotes tissue regeneration on wound surfaces, which is of interest to medicine [96].

## 9. Antibacterial Films from Chitosan with Plant Extracts

Incorporating plant extracts into chitosan films can improve their antibacterial properties and expand their applications. In this section, we will discuss the synthesis and properties of chitosan films with plant extracts and their applications in the food industry and medicine.

### 9.1. Use in the Food Industry

Chitosan films containing plant extracts can be synthesized using methods like solvent casting and electrospinning. Plant extracts, such as essential oils, phenolic compounds, and flavonoids, have been widely investigated for their antimicrobial properties [13].

Chitosan films incorporating plant extracts have been used in food packaging to extend shelf life and maintain quality. For example, chitosan films containing thyme essential oil have been shown to inhibit the growth of spoilage bacteria on fresh-cut fruits and vegetables [97]. Chitosan film with Boldo extract showed great protection against lipid oxidation and inhibition of microorganisms on cheese slices at 4 °C [98]. Extracts of sage and rosemary in chitosan film decrease the percentage of swelling and water vapor permeability values and increase antibacterial activity, and the authors propose to use the resulting films for active food packaging [99]. Extracts of turmeric and green tea increased the antioxidant capacity and antifungal activity of chitosan film used to preserve postharvest strawberries [100]. An edible chitosan film loaded with Nigella sativa L. extract was used to cover grapes. It was shown that the uncovered grapes spoiled, and the grapes covered with films were not spoiled. [101]. Xie et al. prepared a pH-sensitive colorimetric film with red cabbage anthocyanin extract. The elaborated film exhibited good mechanical properties, high barrier ability, excellent thermal stability, significant antioxidant and antimicrobial activity, and an especially sensitive response to pH and ammonia, which can be used to monitor shrimp freshness [102]. A chitosan-based film with poly (vinyl alcohol) and *Phyllanthus reticulatus* fruit extract was prepared in Tilak Gasti’s research group. The obtained new chitosan-based film showed a superior soil degradation rate; this indicates the films are biodegradable in nature. Further, the films demonstrated extensive antimicrobial properties in comparison with the blank chitosan-poly(vinyl alcohol) film. The elaborated film is recommended for application in food biotechnology [103]. Similarly, a chitosan-based film with anthocyanin-rich extract from black rice had strong antioxidant activity, antimicrobial effect, and different color responses to various conditions. The introduction of anthocyanins enables the composite film’s color to change from red to blue with the degree of meat spoilage increased. This circumstance was used as the indicative property of the composite films with regard to meat rotting [104]. Green pea (*Pisum sativum* L.) extract included in chitosan film improved antioxidant and antibacterial activity, tensile strength, and elongation percentage. Moreover, compared to control samples, packing corn oil in green pea pod extract films slowed the rise of thiobarbituric acid and peroxide values [105]. A film based on chitosan and Arabic gum enriched with grape seed extract showed significantly increased thickness and resistance to water vapor while reducing water solubility, visible light transmission, and rotational force in comparison to films without grape seed extract. Cookies with this edible coating were harder and tougher but with much better sensory properties than control during 6 months of storage, and it was confirmed that the coated cookies with the edible film were safe and of acceptable quality after 6 months of storage [106].

### 9.2. Use in Medicine

Plant extracts are widely used in medicine due to their rich pharmacological effects (antimicrobial, anti-inflammatory, antianginal, antispasmodic, regenerative, and much more), which are due to the presence of natural biologically active compounds in these extracts (flavonoids, tannins, saponins, alkaloids, etc.) [107]. Not surprisingly, these circumstances have contributed to the medical studies of polymer films loaded with plant extracts. Among these, chitosan-based films can also be loaded with plant extracts, and this is interesting for biomedical applications.

Ganesan et al. elaborated new attractive chitosan films with *Aloe barbadensis Mill*, *Cissus quadrangularis* and *Curcuma longa* extracts. This film possesses significant antibacterial activity against *Escherichia coli* and *Staphylococcus aureus*. Furthermore, the film had no cytotoxic reactivity to fibroblast cells and test silk fibroin blended film showed slight cytotoxic reactivity to fibroblast cells after 24 h contact [108]. Poly(vinyl alcohol)/Chitosan film loaded by *Basella alba* stem extract is also interesting due to its good antibacterial activity against the foremost infectious bacterial strains, *S. aureus* and *E. coli*. Additionally, *Basella alba* stem extract integrated chitosan/poly(vinyl alcohol) film showed anti-inflammatory properties, hemocompatibility, and excellent biocompatibility, and the in vitro scratch assay and cell adhesion test results illustrated prominent wound healing and adhesion [109]. Chitosan films embedded with bud poplar extract demonstrated good antimicrobial and antibiofilm properties against Gram-positive bacteria and the yeast *Candida albicans*. Bud poplar extract induced an immunomodulatory effect on human macrophages, and this film induced a good regenerative effect in human fibroblasts by in vitro cell migration assay [110].

## 10. Conclusions

This review has provided a comprehensive overview of antibacterial chitosan films and their applications in the food industry and medicine. We have discussed various types of chitosan films, including pure chitosan films, films with metal nanoparticles, films with metal oxide nanoparticles, films with graphene, films with fullerene and its derivatives, and films with plant extracts. We have discussed the properties and potential applications of these films and highlighted the significant role that chitosan films play in extending shelf life and maintaining the quality of food products. In addition, we have discussed their use as wound dressings, drug delivery systems, and medical device coatings in medical applications.

The incorporation of different additives, such as metal nanoparticles, metal oxide nanoparticles, graphene, fullerene derivatives, and plant extracts, can further enhance the antibacterial properties of chitosan films. These additives can help to overcome the limitations of pure chitosan films, such as low solubility and limited antimicrobial activity against specific microorganisms.

The continued development of chitosan films with improved properties will undoubtedly expand their potential applications in various industries. Future research should focus on optimizing the synthesis methods and understanding the structure-property relationships of these materials. Additionally, more studies are needed to evaluate the safety, biocompatibility, and long-term stability of chitosan films in both food and medical applications.

Overall, antibacterial chitosan films hold great promise as a versatile and eco-friendly material for a wide range of applications. Their unique properties, including biodegradability, biocompatibility, and antimicrobial activity, make them an attractive choice for researchers and industry professionals alike. As for the further prospects for the widespread introduction of antimicrobial films of chitosan into clinical practice, the authors of the current review believe that these prospects await chitosan and its derivatives in the near future. This is primarily due to the wide availability of chitosan, a wide variety of its positive biological and pharmacological properties, as well as the convenience of chemical modification, unlike other polysaccharides (due to the presence of primary amine functionality in the macromolecule). The same limitations as the difficulty of standardizing chitosan in the pharmacopoeial analysis will undoubtedly be overcome, and, in the opinion of the authors, they are not a serious obstacle.

## Figures and Tables

**Figure 1 ijms-24-10738-f001:**
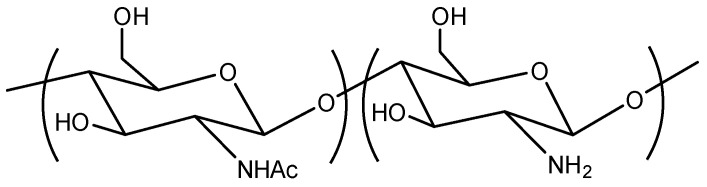
Chemical structure of chitosan.

**Figure 2 ijms-24-10738-f002:**
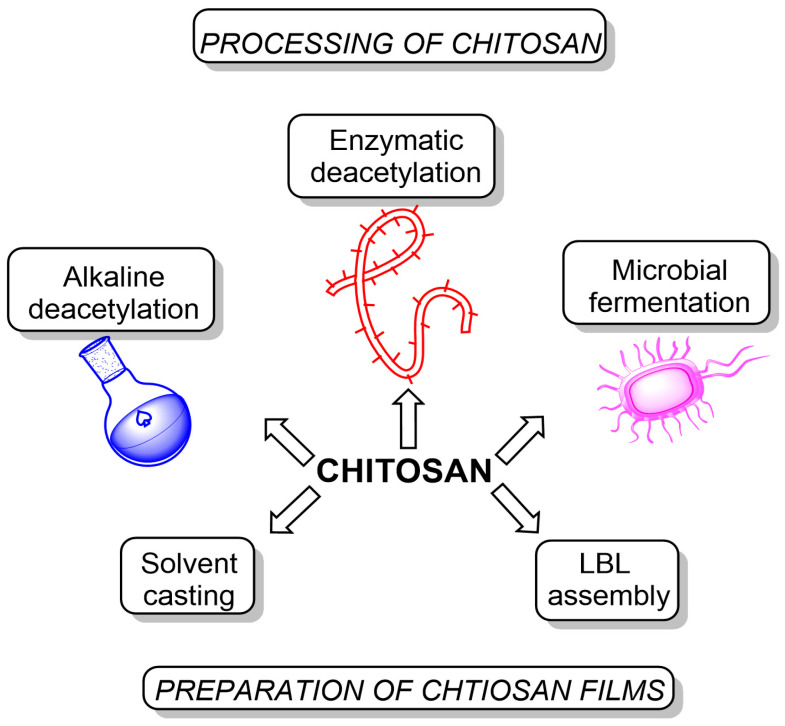
Processing of chitosan and methods of preparation of chitosan-based films.

**Figure 3 ijms-24-10738-f003:**
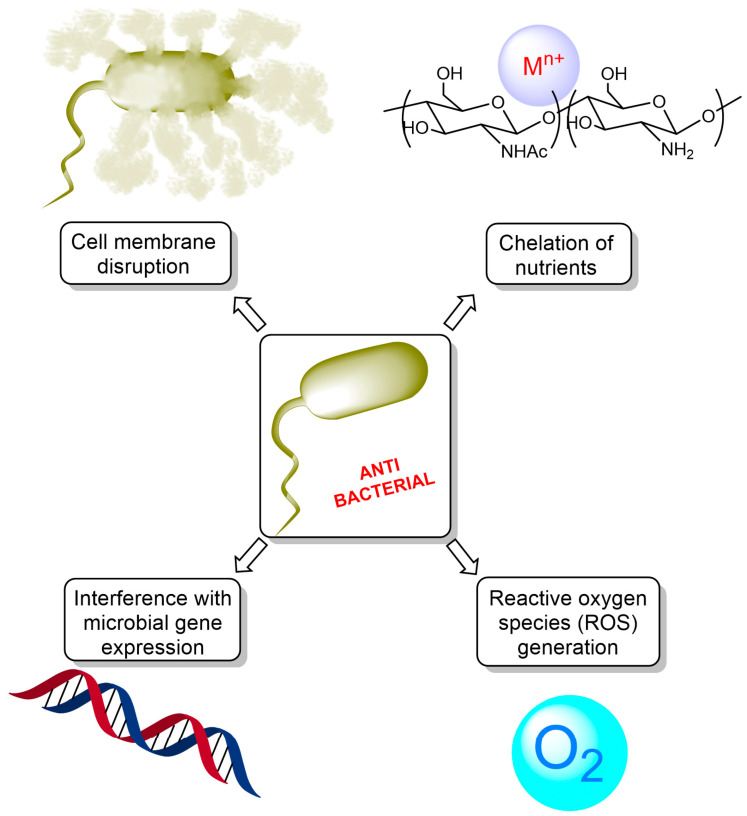
Mechanisms of antibacterial action of chitosan.

## Data Availability

Not applicable.

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
