# Peer review of "Chitosan-Based Antibacterial Films for Biomedical and Food Applications"

_ijms, 2023, doi:10.3390/ijms241310738_

Round 1
Reviewer 1 Report
In this research, the authors reviewed the current development of Chitosan-Based Antibacterial Films for Biomedical and Food Applications. Generally, it’s meaningful and interesting review. In my opinion, the current version of this manuscript fits the scope of IJMS and could be accepted after major revision.
My specific comments are in detail listed below:
1. If possible, a better graphic abstract could be added.
2. In Line 30-37, the usage of chitosan or its derivants to sensitize tumor therapy could be added to better show the merits of chitosan. Some references should be added to this part including 10.1016/j.ijbiomac.2022.10.167.
3. Some minor mistakes exist in the references. The authors should correct it.
4. If possible, the usage of chitosan derivant to form film or other materials could be added.
5. In the review, the possibility of using chitosan or its derivants, including film to cure some other diseases such as tumor could be added. Some references should be added to this part including 10.1016/j.carbpol.2022.119878.
6. A more depth outlook or prospect that pointing out the clinical transformation possibility of the using chitosan or its derivants including film should be added.
Author Response
First, the authors are deeply and sincerely grateful to Reviewers and Editor for their unselfish and extremely important work, for thoroughly checking the manuscript, for valuable comments and advice, which have significantly improved the starting text of the submitted manuscript. Our responses follow below.
We cordially wish you, your family, and your friends and colleagues good health, success and happiness!
- If possible, a better graphic abstract could be added.
- Corrected.
- In Line 30-37, the usage of chitosan or its derivants to sensitize tumor therapy could be added to better show the merits of chitosan. Some references should be added to this part including 10.1016/j.ijbiomac.2022.10.167.
- Corrected.
- Some minor mistakes exist in the references. The authors should correct it.
- Corrected.
- If possible, the usage of chitosan derivant to form film or other materials could be added.
- Corrected.
- In the review, the possibility of using chitosan or its derivants, including film to cure some other diseases such as tumor could be added. Some references should be added to this part including 10.1016/j.carbpol.2022.119878.
- Corrected.
- A more depth outlook or prospect that pointing out the clinical transformation possibility of the using chitosan or its derivants including film should be added.
- Corrected.
Reviewer 2 Report
I read with great interest the paper submitted by Khubiev et al. It is a good list of methods and applications of chitosan. It could be of interest to the readers of this journal. The paper is well-written and clear to understand. The list of references appear to be complete.
However, this paper suffers from the same weakness as many other review papers as it simply provide a list of methods and applications without any critical addition and interpretation. It has been often written for many applications: “… have been widely investigated for their antimicrobial properties”. It would be interesting to know the reasons it works and provide an expert point of view on the subject, which I do not find in this paper.
Author Response
First, the authors are deeply and sincerely grateful to Reviewers and Editor for their unselfish and extremely important work, for thoroughly checking the manuscript, for valuable comments and advice, which have significantly improved the starting text of the submitted manuscript. Our responses follow below.
We cordially wish you, your family, and your friends and colleagues good health, success and happiness!
However, this paper suffers from the same weakness as many other review papers as it simply provide a list of methods and applications without any critical addition and interpretation. It has been often written for many applications: “… have been widely investigated for their antimicrobial properties”. It would be interesting to know the reasons it works and provide an expert point of view on the subject, which I do not find in this paper.
- Corrected. As far as we could it.
Reviewer 3 Report
The authors present a review paper dedicated to the current achievements in the application of chitosan-based films in biomedicine and the food industry. The main synthetic paths towards chitosan derivatization from chitin are pointed out as well as the methods of preparation of chitosan-based films. The very recent applications of antibacterial films of chitosan and chitosan-based materials containing metal or metal oxide nanoparticles, graphene and fullerene or hydroxyl-fullerene are discussed. Finally, chitosan-based films with various plant extracts are discussed. The main emphasis is drawn on the application of these materials in medicine and in food industry. The manuscript is well-written and might be of interest for the readers of International Journal of Molecular Sciences. I have the following questions and remarks:
1. What is the desired degree of chitin deacetylation? It might be useful for the readers if the authors point out the degree of deacetylation achieved by the different methods.
2. The authors should reconsider the claim that the electrospinning is a method for chitosan film formation. I don’t think that the electrospinning is a method for film-formation. First of all, pure chitosan is almost impossible to be electrospun due the its polyelectrolyte nature. Usually, the electrospinning is performed from a mixture with another polymer. The obtained material is a mat and not a film. The reference 24 given by the authors (page 3, line 121) is about electrospinning of zein and chitosan film formation through the solvent casting method. So, it is not appropriate to illustrate the claimed by the authors “electrospinning method for chitosan film formation” by that reference.
3. A number of references from the list are incomplete – page numbers are missing; editors and pages from the cited books and so on. For example: Ref. 2, 4, 11, 14, 21, 36, 38, 44, 52, 62, 87, 89, 96, 105, 107 and 110.
4. Other minor points:
- page 9, line 347: it should be “its” instead of “it”;
- page 11, line 443: it should be “was shown” instead of “was showed”.
Author Response
First, the authors are deeply and sincerely grateful to Reviewers and Editor for their unselfish and extremely important work, for thoroughly checking the manuscript, for valuable comments and advice, which have significantly improved the starting text of the submitted manuscript. Our responses follow below.
We cordially wish you, your family, and your friends and colleagues good health, success and happiness!
- What is the desired degree of chitin deacetylation? It might be useful for the readers if the authors point out the degree of deacetylation achieved by the different methods.
- Corrected
- The authors should reconsider the claim that the electrospinning is a method for chitosan film formation. I don’t think that the electrospinning is a method for film-formation. First of all, pure chitosan is almost impossible to be electrospun due the its polyelectrolyte nature. Usually, the electrospinning is performed from a mixture with another polymer. The obtained material is a mat and not a film. The reference 24 given by the authors (page 3, line 121) is about electrospinning of zein and chitosan film formation through the solvent casting method. So, it is not appropriate to illustrate the claimed by the authors “electrospinning method for chitosan film formation” by that reference.
- Corrected
- A number of references from the list are incomplete – page numbers are missing; editors and pages from the cited books and so on. For example: Ref. 2, 4, 11, 14, 21, 36, 38, 44, 52, 62, 87, 89, 96, 105, 107 and 110.
- Corrected
- Other minor points:
- page 9, line 347: it should be “its” instead of “it”;
- page 11, line 443: it should be “was shown” instead of “was showed”.
- Corrected
Reviewer 4 Report
The general impression from reading the manuscript is that it is clear written. The authors review the most recent articles devoted to the methods used for preparation of chitosan, antimicrobial property of chitosan, types of chitosan-based antimicrobial films and their application in food and medical compositions. Language is good and does not requires any corrections.
Unfortunately, the authors do not describe the specifics of the mechanism of antimicrobial action of chitosan films via adsorption and inactivation of microbial cells on the surface of the films. The description would be desired. However, all previously published reviews have the same drawback.
Conclusion: despite this drawback, I believe that this review can be accepted for publication in the presented form.
Author Response
First, the authors are deeply and sincerely grateful to Reviewers and Editor for their unselfish and extremely important work, for thoroughly checking the manuscript, for valuable comments and advice, which have significantly improved the starting text of the submitted manuscript. Our responses follow below.
We cordially wish you, your family, and your friends and colleagues good health, success and happiness!
Unfortunately, the authors do not describe the specifics of the mechanism of antimicrobial action of chitosan films via adsorption and inactivation of microbial cells on the surface of the films. The description would be desired. However, all previously published reviews have the same drawback.
- Corrected. As far as we could it.
Round 2
Reviewer 1 Report
The current version of this manuscript could be accepted.
Author Response
Thank you!
Reviewer 3 Report
All the questions and issues mentioned in my initial report we properly addressed by the authors in their revised version of the manuscript.
Author Response
Thank you!